# Sitagliptin Induces Tolerogenic Human Dendritic Cells

**DOI:** 10.3390/ijms242316829

**Published:** 2023-11-27

**Authors:** Marija Drakul, Sergej Tomić, Marina Bekić, Dušan Mihajlović, Miloš Vasiljević, Sara Rakočević, Jelena Đokić, Nikola Popović, Dejan Bokonjić, Miodrag Čolić

**Affiliations:** 1Medical Faculty Foca, University of East Sarajevo, 73300 Foča, R. Srpska, Bosnia and Herzegovina; marijadrakul@gmail.com (M.D.); dusan.a.mihajlovic@gmail.com (D.M.); vasiljevicmilos85@gmail.com (M.V.); saradrakocevic@gmail.com (S.R.); dejan.bokonjic@ues.rs.ba (D.B.); 2Institute for the Application of Nuclear Energy, University of Belgrade, 11000 Belgrade, Serbia; sergej.tomic@inep.co.rs (S.T.); marina.bekic@inep.co.rs (M.B.); 3Institute of Molecular Genetics and Genetic Engineering, University of Belgrade, 11000 Belgrade, Serbia; jelena.djokic@imgge.bg.ac.rs (J.Đ.); popovicnikola@imgge.bg.ac.rs (N.P.); 4Serbian Academy of Sciences and Arts, 11000 Belgrade, Serbia

**Keywords:** dipeptidyl peptidase 4 inhibitors, dendritic cells, tolerance, CD26 expression, regulatory T cells

## Abstract

Sitagliptin, an anti-diabetic drug, is a dipeptidyl peptidase (DPP)-4/CD26 inhibitor with additional anti-inflammatory and immunomodulatory properties. In this study, we investigated for the first time the effect of sitagliptin on the differentiation and functions of human dendritic cells generated from monocytes (MoDCs) for 4 days using the standard GM-CSF/IL-4 procedure. LPS/IFN-γ treatment for an additional 24 h was used for maturation induction of MoDCs. Sitagliptin was added at the highest non-cytotoxic concentration (500 µg/mL) either at the beginning (sita 0d protocol) or after MoDC differentiation (sita 4d protocol). Sitagliptin impaired differentiation and maturation of MoDCs as judged with the lower expression of CD40, CD83, CD86, NLRP3, and HLA-DR, retention of CD14 expression, and inhibited production of IL-β, IL-12p70, IL-23, and IL-27. In contrast, the expression of CD26, tolerogenic DC markers (ILT4 and IDO1), and production of immunoregulatory cytokines (IL-10 and TGF-β) were increased. Generally, the sita 0d protocol was more efficient. Sitagliptin-treated MoDCs were poorer allostimulators of T-cells in MoDC/T-cell co-culture and inhibited Th1 and Th17 but augmented Th2 and Treg responses. Tolerogenic properties of sitagliptin-treated MoDCs were additionally confirmed by an increased frequency of CD4+CD25+CD127- FoxP3+ Tregs and Tr1 cells (CD4+IL-10+FoxP3-) in MoDC/T-cell co-culture. The differentiation of IL-10+ and TGF-β+ Tregs depended on the sitagliptin protocol used. A Western blot analysis showed that sitagliptin inhibited p65 expression of NF-kB and p38MAPK during the maturation of MoDCs. In conclusion, sitagliptin induces differentiation of tolerogenic DCs, and the effect is important when considering sitagliptin for treating autoimmune diseases and allotransplant rejection.

## 1. Introduction

Dipeptidyl peptidase-4 (DPP-4), also known as CD26, is a type II transmembrane glycoprotein with a molecular mass of 110 kD. Human DPP-4/CD26 consists of an intracellular domain (composed of six short amino acids), a transmembrane region, and an extracellular domain that possesses DPP activity [1,2,3]. DPP-4/CD26 is widely expressed in various organs (kidney, brain tissue, lung, liver, gastrointestinal tract, and bone marrow), as well as on the surface of various cell types such as endothelial, epithelial, stromal, stem cells, and cells of the immune system (lymphocytes, monocytes, Natural Killer (NK) cells, subsets of dendritic cells, and mast cells) [4,5,6]. There is also a soluble form of DPP-4/CD26 in body fluids, with significant enzymatic activity [7,8]. Entero-endocrine incretins such as glucagon-like peptide 1 (GLP1) and glucose-dependent insulotropic peptide (GIP), known to be important for glucose homeostasis, are rapidly degraded by DPP-4. DPP-4 inhibition slows degradation and increases the half-life of active serum incretins, which then regulate glucose levels [9,10].

DPP-4 inhibitors are a class of relatively new oral anti-hyperglycemic agents indicated for patients with Type 2 Diabetes Mellitus (T2DM) [11,12]. Sitagliptin was the first selective DPP-4 inhibitor approved by the US Food and Drug Administration (FDA) in 2006 for T2DM treatment followed by the development of a number of other selective DPP-4 inhibitors [13,14]. In recent years, many clinical and experimental studies and in vitro research indicate other effects of DPP-4 inhibitors, independently of their enzymatic inhibition and hypoglycemic action [15]. These effects include, among others, suppression of inflammation and modulation of immune functions [16,17]. In this context, the administration of sitagliptin in patients with T2DM reduces the expression of pro-inflammatory markers such as Nuclear Factor kappa B (NF-kB) transcription factor, C-Reactive Protein (CRP), and Interleukin (IL)-6 [16]. In another study on patients with T2DM, sitagliptin inhibited the production of pro-inflammatory cytokines and potentiated the M2 polarization of monocytes [17]. In addition, sitagliptin therapy inhibited monocyte migration into the endothelium of the aorta by a Rac-dependent mechanism using an Apolipoprotein (Apo)E mouse model [18]. Telikani et al. found that sitagliptin in combination with vitamin D3 up-regulated Forkhead box Protein 3 (FoxP3) expression and IL-37 production, and down-regulated Retinoic Acid Receptor (RAR)-Related Orphan Receptor gamma t (RORγt), B-Cell Lymphoma 6 (BCL6), Interferon (IFN)-γ, and IL-17 and IL-21 production, in vivo in patients with T2DM [19]. Pinheiro et al. showed that sitagliptin reduced the levels of T helper (Th)1, Th17, Th2, and IL-6 cytokines in cultures of human peripheral blood mononuclear cells (PBMCs) stimulated with phytohemagglutinin (PHA), simultaneously with increased production of Transforming Growth Factor (TGF)-β, down-regulated expression of CD26, and decreased proliferation of PBMCs [20]. A study on CD26 knock-out mice suggested that CD26 regulates the development, maturation, and migration of CD4+ T cells, NK, and NKT cells and modulates cytokine secretion, T cell-dependent antibody production, and immunoglobulin (Ig) isotype switching in B cells [21].

Dendritic cells (DCs) are the most potent antigen-presenting cells (APCs), specialized for triggering and regulating the immune response by directing the activation and differentiation of T cells. Conversely, DCs are key to maintaining self-tolerance [22,23]. Gliddon et al. [24] first described DPP-4/CD26 expression on DCs from bovine lymph nodes, while Zhong et al. [25] showed increased DPP-4/CD26 expression on DCs from visceral adipose tissue in obese humans and rodents. Schutz and Hackstein showed that CD26 is expressed on the myeloid DC2 subset in human peripheral blood, in contrast to the CD26-negative DC1 subset [26]. In a mixed leukocyte reaction (MLR) assay, CD26 on T cells interacts with Adenosine Deaminase (ADA) on DCs, resulting in stimulation of T-cell proliferation via adenosine degradation [25]. Despite these findings, little is known about whether and how sitagliptin modulates DC-mediated immune responses, and therefore, this was the primary goal of our study. Using a model of human DCs differentiated from monocytes in vitro (MoDCs), we investigated the effect of sitagliptin on differentiation, maturation, allostimulation, Th polarization, and development of subsets of Tregs, including some mechanisms involved.

## 2. Results

### 2.1. Cytotoxicity of Sitagliptin in MoDC Cultures

The first aim of this study was to examine the cytotoxicity of sitagliptin in MoDC cultures. Immature (Im) MoDCs, generated by treating monocytes with DC-differentiating stimuli for 4 days, were treated with double-increasing concentrations of sitagliptin ranging from 31.25 to 1000 µg/mL for 24 h. MTT and apoptosis/necrosis assays were used. MTT showed that only the highest concentration of sitagliptin reduced the metabolic activity (viability) of MoDCs (*p* < 0.01) compared with untreated cultures (Figure 1A). The same results were confirmed in the other assay. As shown in Figure 1B,C, the percentage of early and late apoptotic and necrotic MoDCs was increased only at the highest concentration of sitagliptin (*p* < 0.05).

### 2.2. Sitagliptin Impairs Differentiation and Maturation of MoDCs

Preliminary testing showed that sitagliptin modulates phenotypic and functional properties of MoDCs at 100, 250, and 500 µg/mL, corresponding to about 0.2, 0.5, and 1 mM, respectively (Appendix A). The most pronounced effect was seen with the highest concentration; therefore, the concentration of 500 µg/mL was used in all the next experiments. The differentiation of MoDCs was followed by the almost complete down-regulation of CD14 and up-regulation of CD1a on more than half of MoDCs. When sitagliptin was added at the beginning of monocyte cultures (sita 0d protocol), inhibition in the down-regulation of CD14 was observed (*p* < 0.005) without significant modulation of CD1a expression. The results suggest that sitagliptin present during MoDC differentiation impaired the differentiation process. No changes in the CD1a/CD14 expression occurred when sitagliptin was added after MoDC differentiation (sita 4d protocol) (Figure 2).

Both sitagliptin treatments (sita 0d and sita 4d) inhibited the expression of CD83, CD86, and IL-1β in imMoDCs (*p* < 0.05). Sitagliptin treatment after MoDC differentiation was followed by decreased HLA-DR and CD40 expression in imMoDCs (*p* < 0.05 and *p* < 0.005, respectively), whereas the sita 0d protocol up-regulated HLA-DR expression by these cells (*p* < 0.05) (Figure 3 and Figure 4).

Maturation (m) of MoDC was induced with the treatment of imMoDCs with the combination of Lipopolysaccharide (LPS) and IFN-γ. The maturation was characterized by an up-regulation of HLA-DR, CD86, CD83, IL-1β, and NLRP3 inflammasome compared with imMoDCs. MoDCs matured in the presence of sitagliptin displayed a significantly lower level of these markers. Both sitagliptin treatment protocols inhibited the expression of CD83, CD86, NLRP3, and IL-1β in mMoDCs, and the effect was more pronounced when sitagliptin was applied during MoDCs differentiation (sita 0d protocol). Inhibition of CD40 and HLA-DR by mMoDCs was seen only with the sita 4d protocol (*p* < 0.05). In contrast, HLA-DR was up-regulated (*p* < 0.05) using the sita 0d protocol (Figure 3 and Figure 4).

To assess whether sitagliptin at the tested concentration (500 µg/mL) affects cell viability at the end of the investigated period (5 days), the number of viable MoDCs was calculated (Appendix A). The obtained results showed that sitagliptin did not affect the viability of either im- or mMoDCs, independently of the sita protocol used.

### 2.3. Sitagliptin Up-Regulates the Expression of CD26 on MoDCs

CD26 was non-detected on human monocytes in our initial experiments. Since sitagliptin significantly impaired MoDC differentiation and maturation, we asked whether and how sitagliptin treatment modulates CD26 expression on both imMoDCs and mMoDCs. MoDCs were stained with CD26 and HLA-DR. As shown in Figure 5, the expression of CD26 increased from about 15% in imMoDCs to up to 30% in mMoDCs. Unexpectedly, sitagliptin added during MoDC differentiation up-regulated CD26 expression up to 30% and 55% in imMoDCs (*p* < 0.05) and mMoMDs (*p* < 0.01), respectively. The effect of sitagliptin was not significant when added in cultures of differentiated imMoDCs.

### 2.4. Sitagliptin Modulates Cytokine Production by MoDCs and Inhibits Their Allostimulatory Capacity

Control imMoDCs produced low levels of IL-1β, IL-27, IL-6, IL-10, and TGF-β, whereas Tumor Necrosis Factor-α (TNFα), IL-12p70, and IL-23 were undetectable. The levels of all these cytokines were significantly higher in control mMoDC cultures. Sitagliptin (in both sita 0d and sita 4d protocols) inhibited the production of IL-27 by imMoDCs and augmented IL-10 production by these cells using the sita 0d protocol (*p* < 0.05). The sita 0d protocol significantly inhibited the production of IL-12p70, IL-1β, IL-27, and IL-23 (*p* < 0.05) by mMoDCs, whereas the production of TGF-β was augmented (*p* < 0.05). Sitagliptin added after MoDCs differentiation (sita 4d protocol) inhibited the production of IL-1β and IL-27 by mMoDCs (*p* < 0.05) and augmented IL-10 (*p* < 0.01) and TGF-β (*p* < 0.05) production by these cells (Figure 6A).

The allostimulatory capacity of MoDCs was investigated in the coculture of these cells with purified allogeneic T cells at different MoDC/T-cell ratios. As expected, control mMoDCs exerted higher allostimulatory capability compared with imMoDCs. Sita 0d-treated imMoDCs and mMoDCs inhibited the proliferation of allogeneic T cells at all MoDC/T-cell ratios (*p* < 0.05). In contrast, sita 4d-treated mMoDCs inhibited T-cell proliferation only at a 1:40 mMoDC/T-cell ratio (*p* < 0.05) (Figure 6B).

### 2.5. Sitagliptin Modulates T Helper Polarization Capability of MoDCs

MMoDCs induced a stronger production of Th cytokines (especially IL-17 and IFN-γ) than imMoDCs (Figure 7A). This was confirmed using intracellular staining of cytokines in CD4+ T cells (Figure 7B). Sitagliptin-treated mMoDCs suppressed the production of IFN-γ (*p* < 0.01) and IL-17 (*p* < 0.05), and the effect was stronger when sitagliptin was added during MoDC differentiation (sita 0d protocol). However, the sita 0d protocol stimulated the production of IL-4 by mMoDCs (*p* < 0.01) (Figure 7A,C) and imMoDCs (*p* < 0.05) in the intracellular assay (Figure 7C). These results suggest that sitagliptin stimulated the Th2 polarization capability of MoDCs. Sitagliptin-treated imMoDC inhibited IFN-γ production by T cells, independently of the protocol used (*p* < 0.05) (Figure 7A). However, this effect was not seen at the intracellular level (Figure 7C).

### 2.6. Tolerogenic Properties of MoDCs Treated with Sitagliptin

Suppressed Th1 and Th17 and stimulated Th2 responses in the co-culture of sitagliptin-treated MoDCs suggest that sitagliptin may induce tolerogenic properties of MoDCs. Therefore, we analyzed the expression of tolerogenic markers on MoDCs. Both protocols (sita 0d and sita 4d) increased the expression of Ig-Like Transcript (ILT)4 and Indoleamine 2,3-Dioxygenase 1(IDO1) in mMoDCs (*p* < 0.05). The expression of all three markers (ILT3, ILT4, and IDO1) was increased in imMoDCs using the sita 0d protocol (*p* < 0.05) (Figure 8).

These findings correlated with the induction of Tregs in sitagliptin-treated MoDCs and T cell co-culture. Tregs were identified as CD4+CD127-CD25+FoxP3+ cells. As shown in Figure 9 and Figure 10A, the sita 0d protocol significantly increased the percentage of total Tregs as well as the percentage of Tregs within the CD4+ T-cell population (*p* < 0.05). As shown in Appendix A, the total number of T cells under such culture conditions was not significantly different from the corresponding controls. These results correlated with an increased frequency of CD4+Programmed Death (PD)1+ T cells in the co-culture (Appendix A).

TGF-β is a dominant Treg cytokine, so we analyzed its expression under our culture conditions. As shown in Figure 9 and Figure 10A, both protocols (sita 0d and sita 4d) increased the proportion of TGF-β+ Tregs (*p* < 0.05) independently of whether sitagliptin imMoDCs or mMoDCs were used. These results correlated with increased levels of TGF-β in the co-culture supernatants (*p* < 0.05) (Figure 10B). In addition, the levels of another Treg cytokine, IL-10, were increased only when MoDCs treated with sitagliptin during differentiation (sita 0d protocol) were used (*p* < 0.01).

### 2.7. Induction of Tr1 Cells by Sitagliptin-Treated MoDCs

Tr1 cells were identified as CD4+Foxp3-IL-10+ cells. To exclude possible contamination of Tr1 cells with non-Tregs, the gating procedure included IFN-γ-IL-4-CD4+ T cells. The results presented in Figure 11 show that both imMoDCs and mMoDCs differentiated with sitagliptin significantly increased the proportion of Tr1 cells in the coculture of MoDCs with purified T cells (*p* < 0.05). In contrast, MoDCs treated after their differentiation were without effect. Sitagliptin-treated MoDCs did not change the percentage of CD8+IL-10+ T cells when using the same gating strategy (Figure 11B and Appendix A).

Since Th2 cells are also the source of IL-10, we examined the proportion of IL-4+IL-10+ cells within the CD4+ T cell subset. As shown in Appendix A, MoDCs treated with sitagliptin during differentiation increased the proportion of Th2+IL-10+ T cells.

### 2.8. Sitagliptin Inhibits the Expression of p65 NF-kB and p38 MAPK in MoDCs

The transcription factor NF-kB and Mitogen-Activated Protein Kinase (MAPK) have been found to be critical for the maturation and function of human DCs, and their inhibition results in the induction of tolerogenic DCs [27,28]. The Western blot results presented in Figure 12 show the ratio between phosphorylated (*p*) and total forms of p65 subunit of NF-kB and p38 MAPK, respectively. The treatment of MoDCs with LPS/IFN-γ was followed by activation of both p65 NF-kB (*p* < 0.01) and p38 MAPK (*p* < 0.05) compared with imMoDCs. Sitagliptin treatment inhibited activation of both p65 NF-kB and p38 MAPK in MoDCs induced to mature (*p* < 0.05) without any significant effect on imMoDCs.

## 3. Discussion

DPP4 inhibitors are novel antihyperglycemic drugs approved for treating T2DM [11,12]. Except for their main function related to the inhibition of DPP4 activity and increasing concentration of DPP4 substrates, DPP4 inhibitors exert many other functions including the suppression of inflammation and the immune response, which are not connected with enzyme inhibition [16,17,18,19]. However, the mechanisms involved in these effects are poorly understood. Their effect on the differentiation, maturation, and function of human dendritic cells has not been investigated; therefore, this was a primary objective of this study.

At first, we studied the cytotoxicity of sitagliptin on MoDCs in order to find non-toxic concentrations of the inhibitor for further studies. We showed that MoDCs well tolerated sitagliptin at concentrations up to 500 µg/mL and that relative mild cytotoxicity (determined with MTT and apoptosis/necrosis assays) was observed at the concentration of 1000 µg/mL. Khan et al. recently showed similar results on Vero cell lines, but the cytotoxicity of sitagliptin, as revealed with MTT, was slightly higher (about 50% at 1000 µg/mL) [29]. Similar results were confirmed in glioblastoma (GMB) cell lines [30]. The authors showed that sitagliptin inhibited proliferation and induced apoptosis in GBM cells, starting from 0.5 mM corresponding to 1000 µg/mL, and additionally suppressed self-renewal and stemness of glioma stem cells.

In preliminary experiments, we found that sitagliptin concentrations of 100, 250, and 500 µg/mL were modulatory. Since the effect of 500 µg/mL of sitagliptin was the most pronounced and the effect of lower concentrations was not visible in all assays (see Appendix A), we decided to use the highest concentration in further experiments. One question arose as to whether these in vitro concentrations are relevant because sitagliptin is usually used at the daily dose of 100 mg for T2DM treatment. Pharmacokinetic studies showed that plasmatic concentrations of sitagliptin under such conditions are much lower [31]. However, when considering sitagliptin as an immunomodulatory drug, such as its use to prevent acute graft versus host disease in a clinical allograft transplantation study, much higher doses (600 mg twice daily) are effective in immunosuppression with very mild adverse effects [32]. These results suggest that the concentration of 500 µg/mL (1 mM) of sitagliptin could be relevant for studying immunomodulation.

Our cell culture model was MoDCs. DCs generated in vitro from monocytes are the most explored model to study human DCs because of their low abundance in peripheral blood and difficulties in their isolation. We used the most common protocol to study the differentiation of MoDCs by treating monocytes with Granulocyte-Macrophage Colony Stimulating Factor (GM-CSF) and IL-4 [33]. Although such cells are different from conventional (cDCs) detected with gene expression profiling, they act as APCs in tissue, especially during inflammation. Furthermore, MoDCs are mostly used DCs in vaccination protocols in clinical trials [34]. In most of our papers published to date including the last one [35], we provided evidence that imMoDCs down-regulate CD14 (a main monocyte marker) and acquire CD1a (a DC marker). Most frequently, CD1a is not expressed on all imMoDCs, so CD1a+ and CD1a- DCs are developed, which differ by some functional properties [36]. This was the case in our study. In addition, imMoDCs were characterized by low expression of costimulatory molecules (CD40, CD86), CD83 (a DC maturation marker), and IL-1β. In addition, imMODCs were characterized by lower expression of HLA-DR and NLRP3 inflammasome in comparison with mMoDCs that were treated with maturation-inducing stimuli. These findings are in agreement with already published papers [34,35,36,37,38].

We showed that sitagliptin, when added at the beginning of monocyte differentiation (sita 0d protocol), inhibited the down-regulation of CD14 without significant changes in CD1a expression, such that about 30% of cells were CD14+CD1a+, resembling the phenotype of tissue DCs generated from monocytes under inflammatory conditions [34]. These data suggest that sitagliptin inhibited MoDC differentiation. The effect was not visible when sitagliptin was added after MoDC differentiation (sita 4d protocol), suggesting that under such conditions, the re-expression of CD14 on imMoDCs was not induced. However, sitagliptin treatment after imMODC differentiation was followed by an additional decrease in HLA-DR and costimulatory molecules, suggesting its active role in differentiated MoDCs.

To stimulate maturation, we treated imMoDCs with the combination of LPS and IFN-γ, a protocol that is advantageous over other protocols to generate immunogenic MoDCs with Th1-polarization capability aimed to stimulate the anti-tumor immune response [39,40,41]. Actually, maturation means activation of MoDCs [34], which is followed by up-regulation of CD83, costimulatory molecules, production of IL-12, and pro-inflammatory cytokines [37,38]. Our protocol was optimal for MoDCs to acquire all these properties. Activated MoDCs are able to prime T cells and stimulate their proliferation. Depending on the receptors engaged and signaling pathways involved, mMoDCs stimulate T cells into distinct effectors [34,37]. IL-12 and IFN-γ are important for Th1 polarization, IL-4 is necessary for Th2 and TGF-β/IL-6 and IL-23 for Th17 differentiation and clonal expansion, respectively [42].

We showed that both sitagliptin protocols inhibited maturation (activation) of MoDCs, as judged by decreased CD83, CD86, NLRP3, and IL-1β, and the process was followed by inhibition of their allostimulatory activity. The effect of the sita 4d protocol resulted in a stronger effect since CD40 and HLA-DR were additionally down-regulated. Explaining these differences was not easy because almost nothing is known about the mechanisms involved in these processes.

In this context, the expression of CD26 on MoDCs was very interesting. Our results showed for the first time the expression of CD26 on MoDCs and suggested that the molecule is a MoDC maturation marker, bearing in mind its up-regulation on mMoDCs. Previous results published by Schütz and Hackstein [26] did not show such a finding. The reason is unknown, and possible relevant facts may be associated with the differences in monocyte purification, DC differentiation, maturation induction stimuli, the timing of cell analysis, or differences in anti-CD 26 antibody affinity. However, in our study, sitagliptin up-regulated significantly CD26 expression on both imMoDCs and mMoDCs, which was an unexpected observation. Sitagliptin treatment is usually followed by decreased CD26/DPP-4 expression [16,43]. However, up-regulation of its expression in vivo and in vitro has been published in some studies. For example, in patients on sitagliptin therapy, transient higher CD26 levels in PBMCs were found [44]. Sitagliptin treatment up-regulated CD26 expression on a human embryonic kidney cell line (HEK 293), most probably via an autoregulatory feedback mechanism because DPP4 itself does not affect transcription factors or epigenetics nor does it interfere with nuclear signaling pathways [45]. A similar mechanism could be relevant to our study, but the list of molecules that are the target of sitagliptin treatment in MoDCs is not sufficiently known. Other possible mechanisms may involve compensatory up-regulation, removal of regulatory feedback, off-target effect of the inhibitor, or induction of alternative signaling pathways [46,47,48].

Inhibited activation of MoDCs with sitagliptin is followed by a decrease in Th1, Th17, and an increased Th2 response in MoDC/T-cell co-culture. These results agree with decreased production of IL-12, IL-27 (Th1-inducing cytokines), and IL-23 (a Th17-inducing cytokine). Increased Th2 polarization capacity of sitagliptin-treated MoDCs could result from the classical pathway for Th2 cell induction, which includes IL-12 and Th1 cell downregulation [49]. However, up-regulated Th2 response, seen only with sita 0d protocol, does not align with this hypothesis, suggesting that sitagliptin may activate specific genes in MoDCs favoring Th2 differentiation [50]. Our results also agree with in vitro studies when sitagliptin was used in a model of PHA-stimulated PBMC [20]. However, in that study where lower concentrations of sitagliptin were used than concentrations in our study, Th2 was also down-regulated. Some clinical studies also confirmed that sitagliptin therapy modulated Th responses. For example, a one-year clinical study on Latent Autoimmune Diabetes of Adult (LADA) patients treated with sitagliptin showed a higher percentage of Th2 cells and a lower percentage of Th17 cells in peripheral blood than the control group [51]. In patients with T2DM, the production of IL-17 and IFN-γ significantly decreased in PBMC cultures treated with sitagliptin [52]. Cumulatively, our results suggest that modulation of Th responses is mediated by the influence of sitagliptin on DCs, and such data may be relevant for the treatment of autoimmune diseases in which Th1 and Th17 cells play a key role [53].

Inhibited maturation/activation of MoDCs is accompanied by a shift in immunogenicity toward tolerance, as immature and semi-mature DCs exhibit characteristics of tolerogenic DCs [54]. In addition, DCs can also acquire tolerogenic functions in vivo and in vitro in response to various stimuli such as vitamin D3, corticosteroids, rapamycin or IL-10, TGF-β, and the combination of these cytokines [55]. Tolerogenic DCs express or up-regulate several molecules that are important for tolerogenic functions including PD-L1, ILT3, ILT4, Inducible Costimulator Ligand (ICOS-L), Cytotoxic T Lymphocyte Antigen 4 (CTLA-4), and IDO1 and produce IL-10 and TGF-β [55]. Our results agree with these key findings about the phenotypic and functional profile of sitagliptin-treated MoDCs because these cells significantly up-regulated ILT4 and IDO-1, IL-10, and TGF-β. ILT3 was slightly up-regulated on imMoDCs only when the sita 0d protocol was used. It is known that tolerogenic DCs with high ILT4 expression can anergize CD4+ Th cells and CD8+ cytotoxic T lymphocytes (CTLs), promote the differentiation of different types of Tregs including CD4+CD25+CD45RO+ Tregs, Tr1, and CD8+CD28- suppressive T cells, and produce immunosuppressive cytokines [56].

IDO1 is a tryptophan-metabolizing enzyme that exerts potent immunoregulatory effects when expressed in DCs [57]. Recent experiments in mice showed that the tolerogenic IDO1 pathway is expressed in mature cDC1 but not in cDC2. However, IDO1+ cDC1 induces regulatory cDC2 via Aryl Hydrocarbon Receptor (AhR)-mediated metabolic communication [58]. One of the key functions of tolerogenic DCs is the induction of Tregs. We showed that MoDCs treated with sitagliptin during their differentiation were able to induce both classical Tregs (CD4+CD25+CD127-FoxP3+) and Tr1 cells (CD4+IL-10+FoxP3-) in MoDC/T-cell co-cultures, and these findings correlated with increased levels of IL-10 in co-culture supernatants. The percentage of TGF-β+ Tregs was identified in both the sita 0d and sita 4d protocols, and these results also correlated with the levels of TGF-β in co-culture supernatants. However, it should be stressed that TGF-β in culture supernatants may also originate from conventional T cells and MoDCs themselves.

The differences in MoDC-induced differentiation of Treg subsets and Tr1 cells between the sitagliptin treatment protocols suggest that the presence of sitagliptin during differentiation exerts stronger immunosuppression mediated by both immunoregulatory cytokines. In contrast, TGF-β is additionally involved when sitagliptin activates differentiated MoDCs. Our results are also in accordance with previously published data indicating that IDO1 increased the expression of ILT4 on DCs, which promoted the induction of CD4+CD25+FoxP3+ Tregs [59]. Induction of Tregs by sitagliptin-treated MoDCs agrees with the findings that DPP-4 inhibitors significantly increase the proportion of Tregs in NOD mice [22]. Telikani et al. found that sitagliptin, in combination with vitamin D3, up-regulated FoxP3 expression and IL-37 and downregulated IFN-γ, IL-17, and IL-21 production in vivo in patients with T2DM [26]. Although it was difficult in our experiment settings to simultaneously measure the proliferation of conventional CD4+ T cells and Treg subsets, we are confident that both selective induction of proliferation of T cell subsets and their mutual regulation during the polarization drove the cumulative effects that we described in this paper. Considering all these facts, it can be hypothesized that a down-regulated Th1 response and increased differentiation of Treg subsets induced by sitagliptin via its effect through DCs, although helpful for suppressing autoimmunity, is not desirable for anti-tumor immunity. However, this hypothesis needs additional investigations.

The intriguing question is how sitagliptin induces tolerogenic DCs. The effect is probably not associated with DPP-4/CD26 expression but rather with interference of the drug with signaling molecules. Two of them were investigated (p65 of NF-kB and p38 MAPK), and the obtained results showed the inhibitory effect of sitagliptin on the expression of these transcription factors in MoDCs stimulated with maturation-inducing stimuli. Both the NF-kB and MAPK signaling pathways are involved in different functions of DCs, including the production of pro-inflammatory cytokines and up-regulation genes for costimulatory molecules upon LPS stimulation [60,61,62]. These results are in accordance with previous studies showing that sitagliptin inhibited LPS-induced nuclear translocation of NF-kB, p38 MAPK phosphorylation, and subsequent production of IL-6 in Human Umbilical Vein Endothelial Cells (HUVECs) [63]. In addition, the pretreatment of imMoDCs with a p38 MAPK inhibitor SB203580 prevented up-regulation of maturation/activation markers (CD40, CD86, CD80, HLA-DR, and CD83) induced by LPS and TNFα. Additionally, the inhibitor decreased the production of proinflammatory cytokines (IL-12, IL-6, IL-1, and TNFα) by MoDCs in such cultures [64,65]. It remains to be investigated whether CD26 ligands expressed by DCs (ADA, Caveolin 1, and CD45) [25,66,67] are involved in these processes through homotypic interactions of CD26 on MoDC with its ligands on the neighboring MoDC and how sitagliptin interferes with these processes.

In conclusion, our results showed for the first time that sitagliptin induces the development of tolerogenic MoDCs in vitro with the potential to inhibit alloreactivity, suppress Th1 and Th17 immune responses, enhance Th2 response, and stimulate differentiation of Tregs and Tr1 cells. The inhibited maturation of MoDCs was followed by inhibited NF-kB and p38MAPK pathways but, unexpectedly, enhanced CD26 expression. In this context, the obtained results support the clinical findings about the beneficial effects of sitagliptin therapy in autoimmune diseases and undesired effects in the progression of some malignant tumors [68]. This study has some limitations because time and dose-dependent effects of sitagliptin were not analyzed, except for at the preliminary level. The applied concentration of sitagliptin (500 µg/mL), although non-cytotoxic, was relatively high, and its relationship with high doses of sitagliptin used for the prevention of allotransplantation rejection deserves further analysis and research. In addition, the possible mechanisms underlying the tolerogenic properties of MoDCs were not further explored, including the relevance of the increased expression of CD26 by sitagliptin-treated MoDCs in inducing Tregs. However, the obtained results open a perspective for further research, especially for using sitagliptin with the existing and novel immunotherapy protocols.

## 4. Materials and Methods

### 4.1. Generation of MoDCs

All experiment protocols were approved by the Ethical Board of the University of East Sarajevo, Medical Faculty Foča, and carried out following Institutional guidelines. PBMCs were obtained from the buffy coats of healthy volunteers who signed an Informed Consent form by the Declaration of Helsinki, using density gradient centrifugation on lymphocyte separation medium 1077 (PAA, Linz, Austria). CD14+ monocytes and CD3+ T cells were isolated from PBMC with magnetic-activated cell sorting (MACS) using a Monocyte Isolation Kit II and a Pan T cell Isolation Kit (Miltenyi Biotec, Bergisch Gladbach, Germany), respectively, following the instructions of the manufacturer.

To generate imMoDCs, monocytes were cultivated for four days in a GMP Dendritic Cell medium (CellGenix, Freiburg, Germany) supplemented with 100 ng/mL of human recombinant GM-CSF (Novartis, Basel, Switzerland) and 20 ng/mL of human recombinant IL-4 (Roche Diagnostics, Basel, Switzerland) in 6-well plates (Sarstedt, Numbrecht, Germany) at a density of 0.5 × 10^6^/mL, volume 2 mL. The cultures were maintained in a cell incubator (37 °C, 90% humidity, and 5% of CO_2_). To assess the effects of sitagliptin (Sigma-Aldrich, Darmstadt, Germany) on MoDCs, monocytes were treated with 500 µg/mL of sitagliptin (500 µg/mL in most experiments), referred to as the sita 0d protocol. On day 3, the cell culture medium in all cultures was refreshed by adding 500 µL of medium supplemented with GM-CSF and IL-4, with added sitagliptin for the sita 0d cultures. Control imMoDCs were without sitagliptin. After four days of cultivation, imMoDCs were stimulated with 50 ng/mL of IFN-γ (R&D Systems, Minneapolis, MN, USA) and 200 ng/mL of LPS from Escherichia coli 0.111:B4 (Sigma-Aldrich, Darmstadt, Germany) for an additional 24 h to induce cell maturation (mMoDCs). The maturation-induction stimuli were added directly to MoDC cultures. A parallel set of imMoDC cultures was treated with sitagliptin after MoDC differentiation and referred to as the sita 4d protocol. The cultures were preincubated with sitagliptin for 3 h before the addition of LPS/IFN-γ to induce mMoDCs. Sitagliptin was maintained in cultures during the whole culture period. On day 5, MoDCs (control and sitagliptin 0d/4d-treated im- and mMoDCs) were harvested, washed twice with RPMI-1640 medium (Sigma-Aldrich, Darmstadt, Germany), counted, and used in the subsequent assays. The supernatants were collected and frozen at −80 °C until cytokine levels were measured.

### 4.2. Cytotoxicity Assays

Cytotoxicity of sitagliptin was determined using MTT and an apoptosis/necrosis assay. ImMoDCs (1 × 10^5^/well), obtained with a 4-day differentiation period, were cultivated in the complete culture medium in 96 flat-bottom wells (Sarstedt, Numbrecht, Germany) for 24 h. The complete culture medium was RPMI-1640 containing 10% fetal calf serum (FCS), 50 µM 2- mercaptoethanol, and antibiotics (penicillin and streptomycin, 100U/mL (all from Sigma-Aldrich, Darmstadt, Germany). The cells were treated with double decreasing concentrations of sitagliptin starting from 1000 µg/mL to 31.25 µg/mL (MTT assay) or 1000 µg/mL to 62.5 µg/mL (apoptosis/necrosis assay). Untreated imMoDC cultures served as controls. All cell cultures were set up in triplicates. In the MTT assay, imMoDCs were treated with MTT (3-(4,5-dimethylthiazol-2-yl)-2,5-diphenyltetrazolium bromide) for 4 h at a final concentration of 0.5 mg/mL. Blank controls were cell-free cultures with corresponding sitagliptin concentrations. To dissolve the formazan crystals, the samples were incubated overnight in 10% (*w*/*v*) sodium dodecyl sulfate (SDS, Millipore, Burlington, MA, USA) and 0.01 N (*v*/*v*) hydrochloric acid (HCl, Sigma-Aldrich, Darmstadt, Germany). The absorbance was determined at wavelengths of 670 and 570 nm using an ELx800 microplate reader (Biotek in Winooski, Winooski, VT, USA). The optical density (OD) was corrected by subtracting OD 670 from OD 570. The relative metabolic activity (MTT%) in sitagliptin-treated cultures was calculated by subtracting the corrected OD value from the corresponding blank controls and normalizing the results to control imMoDCs (100%).

To determine the mode of cell death (apoptosis and/or necrosis) in MoDC/sitagliptin cultures, the cells were harvested after 24 h. Apoptosis and necrosis were determined using Annexin-V/Propidium Iodide (PI) staining kit (R&D Systems, Minneapolis, MN) following the manufacturer’s instructions. The analysis was performed on a BD LSR II flow cytometer (BD Biosciences, Franklin Lakes, NJ, USA). Signal overlaps between the channels were compensated before each experiment using single-labeled cells. The acquired data were analyzed offline in the FlowJoVX program (BD Biosciences, Franklin Lakes, NJ, USA). Single Annexin-fluorescein isothiocyanate (FITC)-positive cells were recognized as early apoptotic cells, double-positive cells were late apoptotic cells, whereas single PI-positive cells were necrotic cells. Results were expressed in %.

The percentage of dead cells in both im- and mMoDCs cultures after the whole cultivation period (5 days) was determined manually using a light microscope after staining the cells with a solution of Trypan blue (1%). Total viability (% of survival cells in culture relative to the initial number of cells used as 100%) was determined as follows: number of viable cells in culture (5 days)/number of initial cells at the beginning of culture × 100.

### 4.3. Mixed Leukocyte Reaction

An allogeneic Mixed Leukocyte Reaction (MLR) assay was used to evaluate the capability of control and sitagliptin-treated MoDCs to stimulate the proliferation of allogeneic T cells. MoDCs (1 × 10^4^–0.25 × 10^4^/well) were co-cultured with allogeneic (CD3+) T cells (1 × 10^5^/well), purified with immunomagnetic sorting and pre-labeled with CellTrace Far Red (CTFR) dye (Invitrogen, Waltham, MA, USA) in round-bottom 96-wells plates, thus providing 1:10–1:40 MoDC: T-cell ratios, respectively. The complete culture RPMI medium was used. After a cocultivation period of 4 days, the cells were harvested and stained with 7-AAD (50 μg/mL) (Sigma-Aldrich, Darmstadt, Germany) to include only viable cells in the analysis. Proliferation was determined using a flow cytometer (BD Biosciences, Franklin Lakes, NJ, USA) according to the CTFR dilution after excluding doublets. The acquired data were analyzed offline in the FlowJoVX program (BD Biosciences, Franklin Lakes, NJ, USA). To determine cytokine levels in supernatants and intracellular production of cytokines, these co-cultures were incubated with phorbol 12-myristate 13-acetate (PMA, 20 ng/mL) and ionomycin (500 ng/mL) (both from Sigma-Aldrich, Darmstadt, Germany) for 6 h followed by monensin treatment (2 µM, Sigma-Aldrich, Darmstadt, Germany) for 4 h before harvesting the cell-free supernatants and cells.

### 4.4. Th Polarization and Tregs Induction

For Th polarization, MoDC/T cell co-cultures were carried out in 1:20 MoDC: T cell ratio for 5 days in round-bottom 96-wells plates using complete culture medium and then treated with PMA/ionomycin (as described above) for the last 6 h before harvesting the cell-free supernatants for cytokines quantification. To detect intracellular cytokines in T cells, the same co-cultures were treated with PMA/ionomycin and monensin (2 μM) for the last 4 h and then prepared for flow cytometry.

For the induction of Tregs, MoDC/T cell co-cultures were set up in the suboptimal (1:50) MoDC: T cell ratio in the presence of a low dose of human recombinant IL-2 (2 ng/mL, R&D Systems, Minneapolis, MN) for 6 days. The co-cultures were then treated with PMA/ionomycin and monensin, as described above, and prepared for the flow cytometry analysis.

A schematic presentation of the whole process of differentiation and maturation of MoDCs (control, sita 0d, and sita 4d protocols) and MoDCs/T-cell co-culture protocols is given in Appendix A.

### 4.5. Cytokine Measurement

Supernatants from MoDC cultures were assayed for IL-12p17, IL-23, IL-27, IL-6, IL-10, IL-1β, and TGF-β, whereas MoDC/T cell co-cultures were analyzed for IL-4, IL-10, IL-17, IFN-γ, and TGF-β, using specific sandwich enzyme-linked immunosorbent assays (ELISA) (all kits were from R&D Systems, Minneapolis, MN) following the instruction of the manufacturer. Cytokine levels were quantified according to the standard curves established with known concentrations of the cytokines. Each cytokine level was normalized to the standard number of cells calculated after harvesting (5 × 10^5^ MoDCs in MoDC cultures; 1 × 10^5^ T cells in MoDC/T cell co-cultures).

### 4.6. Flow Cytometry

The phenotype analysis of MoDC and T cells was carried out using flow cytometry after staining the cells with fluorescently labeled Abs (Clone) and reagents: IgG1 negative control-phycoerythrin (PE) (MCA928PE) and IgG1 negative control-fluorescein isothiocyanate (FITC) (MCA928F) (Bio-Rad Laboratories, Hercules, CA, USA); anti-CD1a- Peridinin-Chlorophyll-Protein (PerCP)/Cyanine (Cy) 5.5 (HI149), anti-HLA-DR-Allophycocyanin (APC)/Cy7 (L234), anti-IL-4-PerCP/Cy5.5 (MP4-25D2), anti-IL-4-Phycoerythrine (PE) (MP4-25D2), anti-ILT-4-APC (42D1), anti-CD25-PE (BC96), anti-CD25-PerCP/Cy5.5 (M-A251), anti-CD127-PE (A019D5), anti-IL-10-APC, anti-IL-10-PE (JES5-16E3), anti-TGF-β-APC (TW4-6H10), anti-IL17A-Alexa Fluor 488 (BL168), anti-IFN-γ-APC, anti-IFN-γ-FITC (4S.B3), anti-CD83-FITC (HB15e), IgG1 negative control-PerCP/Cy5.5 (HTK888), anti-PD1L-PE (29E.2A3), and anti-CD26-FITC (all from BioLegend, Basel, Switzerland); anti-HLA-DR-PerCP (L243), anti-IDO-1-APC (700838), anti-CD4-FITC, anti-CD4-APC (11830), anti-TGF-β-PE (9016) (all from R&D Systems, Minnesota, USA), anti-CD14-FITC (TUK4), anti-NLRP3-APC (rea668), anti-IL-1β-PE (rea1172) (all from Miltenyi Biotec, Gladbach, Germany); anti-CD86-PE (IT2.2), streptavidin-PerCP, streptavidin APC, anti-ILT3-PE (ZM4.1), IgG1 negative control APC (MA5-18093), and anti-IL-17A-APC (eBio17B7) (all from Thermo Fisher Scientific, Dreieich, Germany); and anti CD40-APC (5C3), anti-IL-12 (p40/p70)-PE (C11.5), anti-CD3-PE (SK7), anti-FoxP3-PerCP/Cy5.5, anti-FoxP3-Alexa Fluor 488 (236A/E7), anti-RORγt-Alexa Fluor 488 (Q21-559), (all from BD Biosciences, San Diego, CA, USA), and anti-CD8-PerCP/Cy5.5 (HIT8a) (Elabscience, Houston, TX, USA). The incubation lasted for 30 min at 4 °C. Intracellular staining was carried out after surface labeling with the fixation and permeabilization kit (BioLegend, Basel, Switzerland). The analysis was performed using a BD LSR II flow cytometer (BD Biosciences, Franklin Lakes, NJ, USA). For each analysis, doublets were excluded according to forward scatter (FSC)-A/FSC-H, and more than 5000 cells were gated according to their specific FSC-A/side-scatter (SSC)-A properties. The analysis excluded cells with low FSC-A/SSC-A properties, representing predominantly dead cells. Signal overlap between the channels was compensated before each experiment using single fluorescence-labeled cells, and the non-specific fluorescence was determined using appropriate isotype control antibodies and fluorescence minus one (FMO) controls. The acquired data were analyzed offline in the FlowJoVX program (BD Biosciences, Franklin Lakes, NJ, USA).

### 4.7. Western Blot

ImMoDCs were treated with sitagliptin (500 µg/mL) for one hour. After that, half of such cultures were treated with LPS/IFN-γ for an additional two hours. Sitagliptin-untreated MoDCs were used as controls. The cells (approximately 2 × 10^5^ MoDCs per sample), were then collected and subjected to protein isolation according to the protocol described by Herholz et al. [69]. Protein concentration was measured with the BCA Assay Kit (Thermo Fisher Scientific, Dreieich, Germany), and 20 μg of extracted proteins as well as the protein ladder (BlueStar Prestained Protein Marker, MWP03) were separated on 12% SDS–PAGE and transferred to a 0.2 mm nitrocellulose membrane (GE Healthcare, Chicago, IL, USA). According to the appropriate ladder bands, the membrane was divided into strips spanning regions around the targeted protein, between 53 kD and 93 kD for p65/pp65, and between 23 kD and 41 kD for p38/pp38. Western blotting was performed overnight at 4 °C with antibodies against: p38 MAPK (D13E1) (1:1000, Cell Signaling, Danver, MA, USA, #8690), phospho-p38 MAPK (Thr180/Tyr182) (1:1000, Cell Signaling, Danver, MA, USA, #9211), NFκB p65 (C-20): sc-372 (1:1000, Santa Cruz Biotechnology, Inc., Dallas, TX, USA), and Phospho-NF-κB p65 (pp65) (Ser536) (93H1) (1:1000, Cell Signaling, Danver, MA, USA, Rabbit mAb #3033). Anti-glyceraldehide-3-phosphate dehydrogenase (GAPDH) rabbit polyclonal antibody (Abcam, Cambridge, UK)) was used as a loading control. Chemiluminescence was detected using the ChemiDoc Touch Imaging System with Image Lab Touch Software 6.1 (Bio-Rad, Hercules, CA, USA). The Image Lab Touch software accurately estimates the shortest exposure time. The Image Resolution/Sensitivity scale was set to 4 × 4-pixel binning settings, and pictures were captured using the Rapid Auto Exposure mode. The density of the bands was quantified in ImageJ 1.50f (National Institutes of Health, NIH, Bethesda, MD, USA) software using plot lanes and the wand tool. Relative band density ratios of pp65/p65 and pp38/p38 were used as a measure of p65 and p38 activation. This is expressed as fold changes of pp65 (pp38) in comparison to total p65 (p38) in each sample [62]. For each condition, three independent replicates were used. Two independent experiments were performed.

### 4.8. Statistical Analysis

The results are presented as representative data or as mean ± SD values of three independent experiments. The normality of the data was tested using the Shapiro–Wilk test. Statistical analyses were performed using Graph Pad Prism software 8.1 (GraphPad, La Jolla, CA, USA). The differences in mean values between control and experimental cultures (normally distributed samples) were analyzed using an unpaired *t*-test with Welch correction. Otherwise, the Mann–Whitney test was used for the comparisons. The Friedman test (paired one-way ANOVA) with Dunn’s multiple comparison post-test was used for comparison, where more than two groups with paired samples were analyzed. Values at *p* < 0.05 or less were considered to be statistically significant.

## Figures and Tables

**Figure 1 ijms-24-16829-f001:**
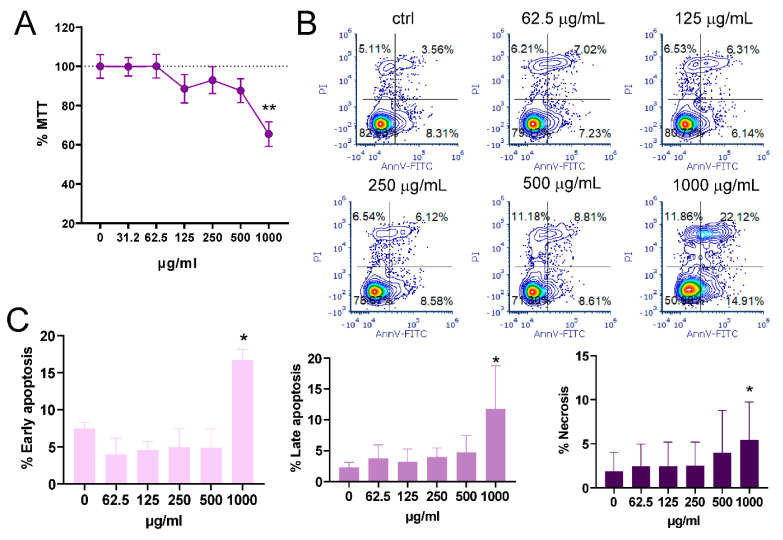
The effect of different concentrations of sitagliptin on the metabolic activity (**A**) and apoptosis/necrosis (**B**,**C**) in immature MoDC cultures. Immature (Im) MoDCs were generated by incubating human monocytes for 5 days in culture medium supplemented with Granulocyte Macrophage-Colony Stimulating Factor (GM-CSF/IL-4), as described in the Materials and Methods. ImMoDCs were treated with double-increasing concentrations of sitagliptin starting from 31.25 to 1000 µg/mL for 24 h. Cytotoxicity was evaluated using MTT and apoptosis/necrosis assays. Values are given as mean ± SD (*n* = 3). * *p* < 0.05, ** *p* < 0.01 compared with corresponding controls (non-treated imMoDCs). (**B**) Plots of apoptosis/necrosis of one representative experiment.

**Figure 2 ijms-24-16829-f002:**
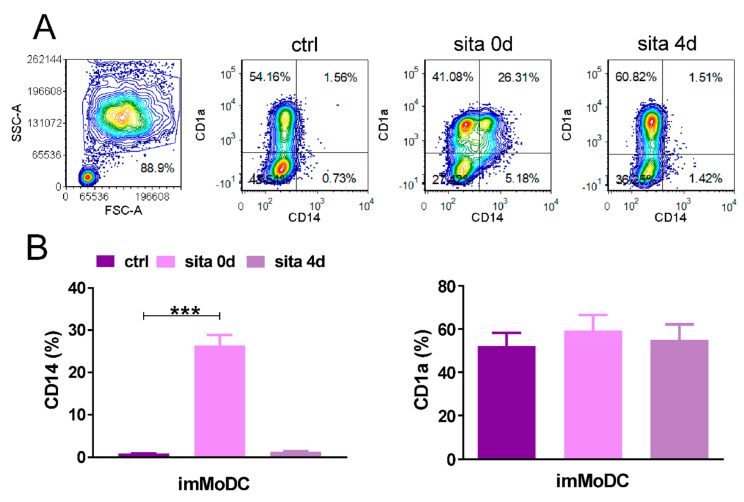
The effect of sitagliptin (500 µg/mL) on the expression of CD1a and CD14 on imMoDCs. Sitagliptin was applied at the beginning of Mo differentiation (sita 0d) or after imMoDC differentiation (sita 4d). CD1a/CD14 expression was analyzed on day 5 of cell cultures. (**A**) Plots of CD1a/CD14 expression of one representative experiment. (**B**) Results are presented as mean ± SD (*n* = 3). *** *p* < 0.005 compared with corresponding controls (non-treated imMoDCs).

**Figure 3 ijms-24-16829-f003:**
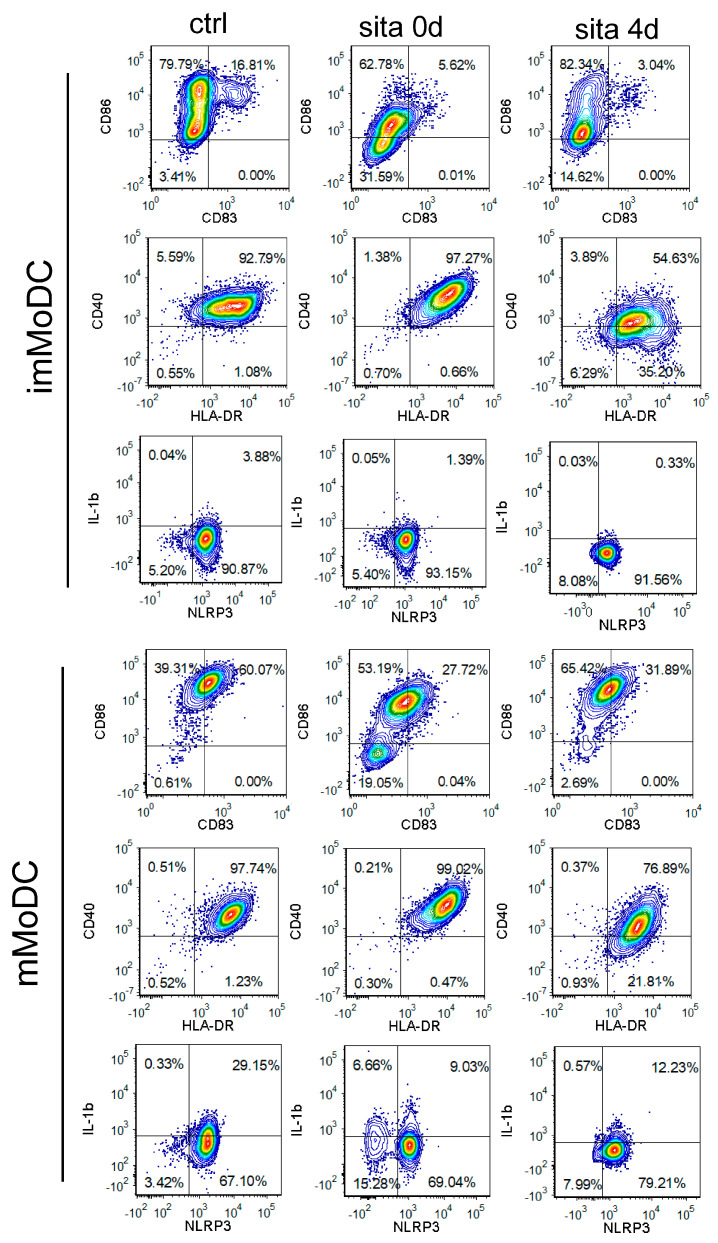
The effect of sitagliptin on the expression of different markers by im- and mMoDCs. Sitagliptin was applied at the beginning of Mo differentiation (sita 0d) or after imMoDC differentiation (sita 4d) at the concentration of 500 µg/mL. The plots are for one representative experiment.

**Figure 4 ijms-24-16829-f004:**
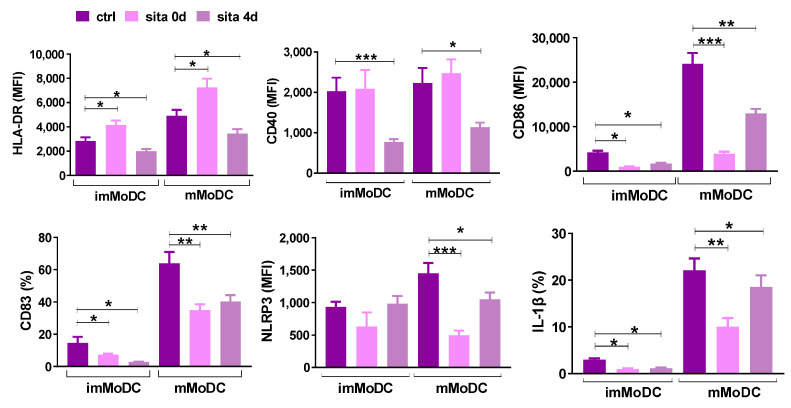
The effect of sitagliptin on the expression of different markers by im- and mMoDCs. Sitagliptin was applied at the beginning of Mo differentiation (sita 0d) or after imMoDC differentiation (sita 4d) at the concentration of 500 µg/mL. Results are presented as mean ± SD (*n* = 3). * *p* < 0.05; ** *p* < 0.01; *** *p* < 0.005 compared with corresponding controls (non-treated MoDCs).

**Figure 5 ijms-24-16829-f005:**
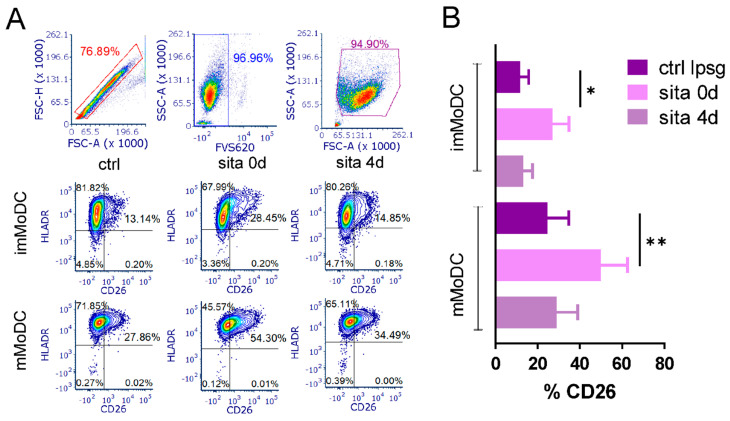
The effect of sitagliptin on the expression of CD26 by MoDCs. Sitagliptin (500 µg/mL) was applied at the beginning of Mo differentiation (sita 0d) or after imMoDC differentiation (sita 4d). CD26/HLA-DR expression was analyzed on day 5 of cell cultures. (**A**) The plots of marker expression by im- and mMoDCs for one representative experiment are shown. (**B**) The results are presented as mean ± SD (*n* = 3). * *p* < 0.05; ** *p* < 0.01; compared with corresponding controls (non-treated imMoDCs).

**Figure 6 ijms-24-16829-f006:**
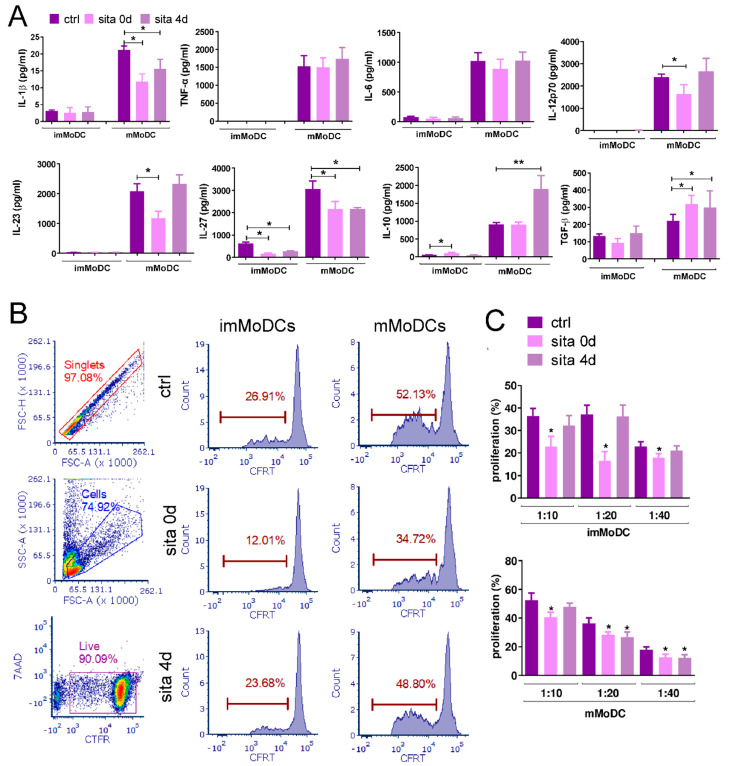
The effect of sitagliptin (500 µg/mL) on cytokine production by MoDCs and their allostimulatory capacity. Day-4 imMoDCs were induced to mature for the next 24 h, as described in the Materials and Methods. The levels of cytokine in supernatants, standardized to an equal number of MoDCs (5 × 10^5^) of both im- and mMoDCs (**A**) were analyzed on day 5 of cell cultivation. Day-5 im- and mMoDCs were co-cultivated with purified CellTrace Far Red-stained T cells for 4 days, and after that, the proliferation was analyzed using flow cytometry (**B**,**C**). 7-Aminoactinomycin D (7-AAD) was used to detect viable cells. Results are presented as mean ± SD (*n* = 3). * *p* < 0.05; ** *p* < 0.01; compared with corresponding controls (non-treated MoDCs). (**B**) Plots of allogeneic T-cell proliferation of one representative experiment.

**Figure 7 ijms-24-16829-f007:**
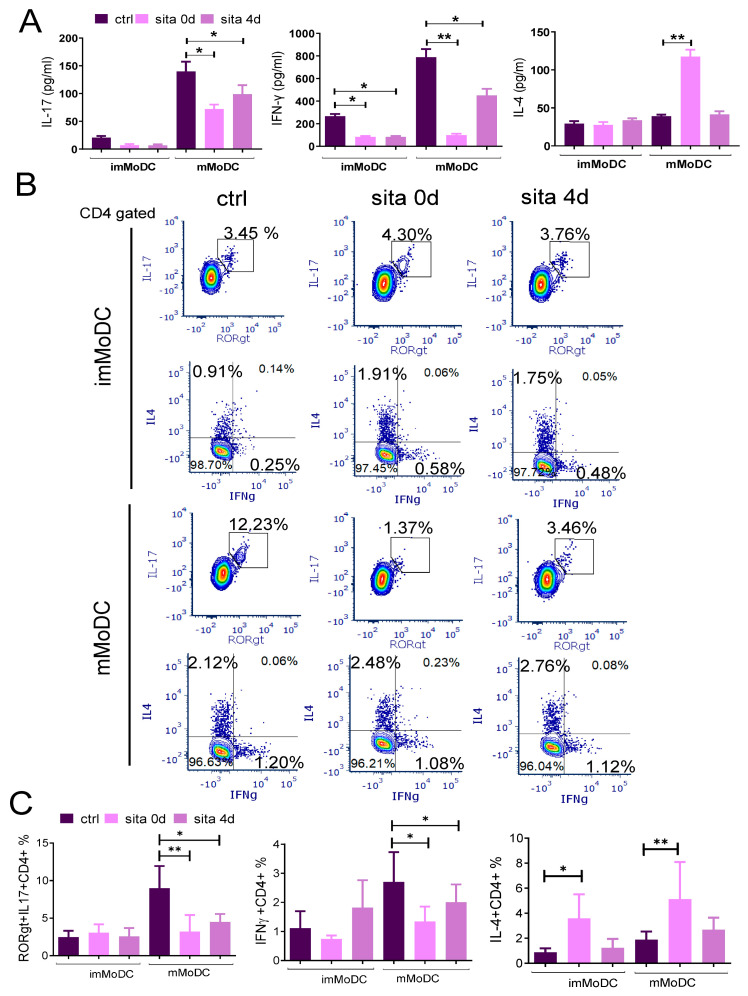
The effect of MoDCs treated with sitagliptin (500 µg/mL) on their Th polarization capability. ImMoDCs and mMoDCs were co-cultivated with purified allogeneic T cells for 4 days. (**A**) The levels of Th cytokines were determined in culture supernatants (standardized to an equal number of cells (1 × 10^5^ T cells) and intracellularly (**B**,**C**), as described in the Materials and Methods. Results are presented as mean ± SD (*n* = 3). * *p* < 0.05; ** *p* < 0.01; compared with corresponding controls (non-treated MoDCs). (**B**) shows the plots of Th cytokine expression of one representative experiment.

**Figure 8 ijms-24-16829-f008:**
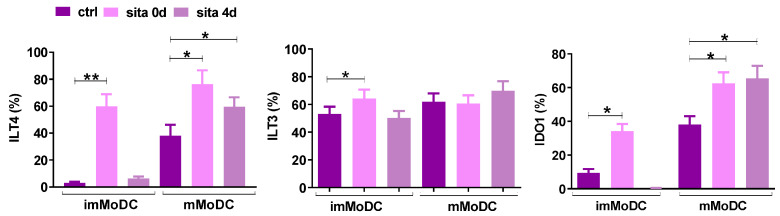
The expression of tolerogenic markers by control and sitagliptin-treated MoDCs. Sitagliptin was used at a concentration of 500 µg/mL. The expression of ILT3, ILT4, and IDO-1 was analyzed on day 5 MoDCs. Results are presented as mean ± SD (*n* = 3). * *p* < 0.05; ** *p* < 0.01; compared with corresponding controls (non-treated MoDCs).

**Figure 9 ijms-24-16829-f009:**
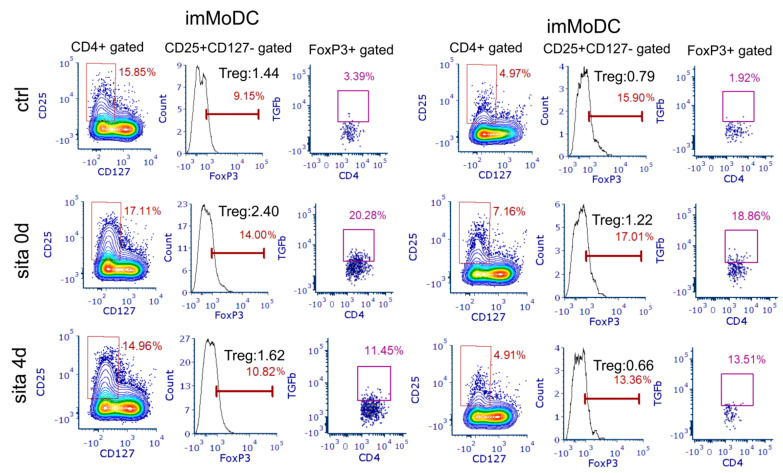
Tolerogenic properties of MoDCs treated with sitagliptin (500 µg/mL). ImMoDCs and mMoDCs were co-cultivated with purified allogeneic T cells for 4 days. Tregs were identified as FoxP3+ cells in CD4+CD25+CD127- cells. The numbers of cells marked in red on flow cytometric plots (middle vertical rows) represent the percentage of CD25+CD127-FoxP3+ cells, whereas the numbers marked in black on the same histograms represent the percentages of Tregs within total CD4+ T cells. TGF-β+ Tregs were identified within CD4+CD25+CD127-FoxP3+ cells and expressed as %. The plots and histograms are from one of three different experiments.

**Figure 10 ijms-24-16829-f010:**
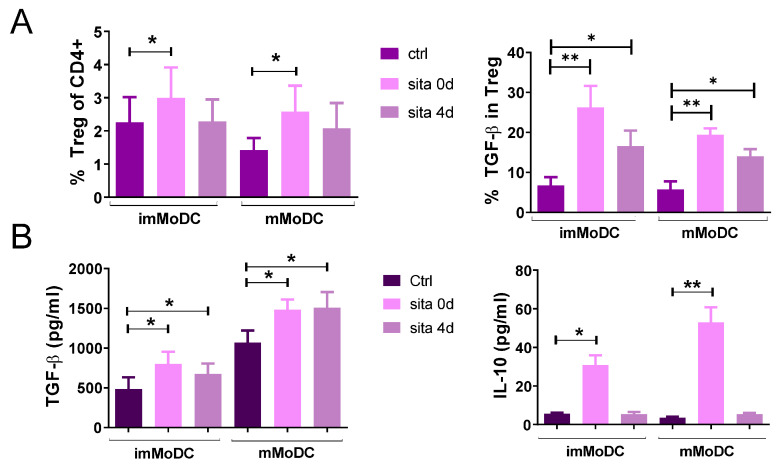
The subsets of Tregs and levels of tolerogenic cytokines (TGF-β and IL-10) in co-culture supernatants. Control MoDCs and MoDCs treated with sitagliptin (500 µg/mL) were co-cultured with purified allogeneic T cells, as described in the Materials and Methods. (**A**) Tregs were identified as FoxP3+ cells in CD4+CD25+CD127- cells and presented in %. The percentages of TGF-β+ Tregs were expressed as relative values (%) of the cells calculated on the basis of the number of total Tregs used as 100%. (**B**) The levels of IL-10 and TGF-β were determined in MoDC/T-cell co-culture supernatants and standardized to an equal number of cells (1 × 10^5^ T cells). Results are presented as mean ± SD (*n* = 3). * *p* < 0.05; ** *p* < 0.01; compared with corresponding controls (non-treated MoDCs).

**Figure 11 ijms-24-16829-f011:**
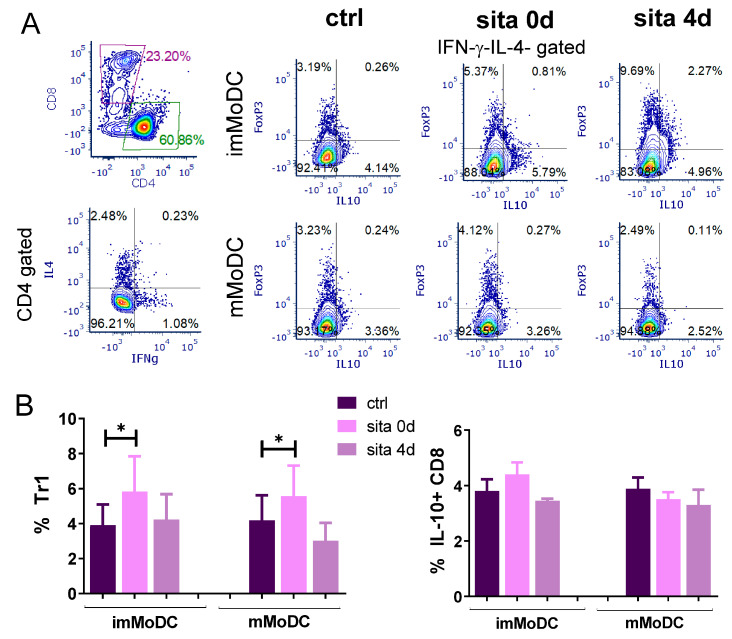
The effect of MoDCs treated with sitagliptin (500 µg/mL) on the induction of Tr1 cells. Tr1 cells were identified as CD4+Foxp3-IL-10+ T cells within the IFN-γ-IL-4-CD4+ T-cell population. (**A**) Plots of one experiment are shown. (**B**) The results are presented as mean ± SD (*n* = 3). * *p* < 0.05 compared with corresponding controls (non-treated MoDCs).

**Figure 12 ijms-24-16829-f012:**
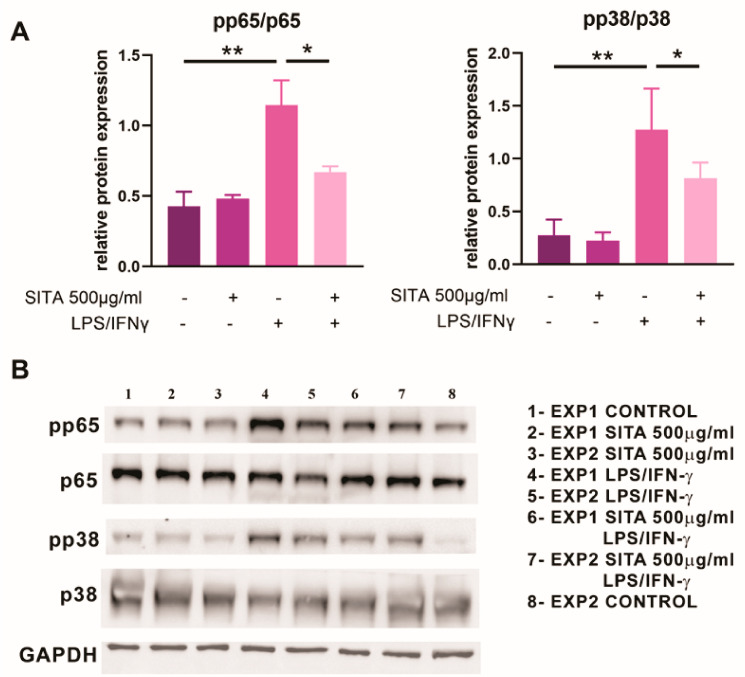
The effect of sitagliptin (500 µg/mL) on the activation of p65 NF-kB and p38 MAPK in MoDCs. The activation of p65 NF-kB and p38 MAPK was analyzed using Western blot by the phosphorylated (*p*) and total forms of these molecules, as described in the Materials and Methods. Values are from two different experiments (**A**); supported by the original blots (**B**). * *p* < 0.05; ** *p* < 0.01 compared with corresponding controls as indicated by bars. GAPDH (Glyceraldehyde-3-phosphate dehydrogenase) was used as a loading control.

## Data Availability

All data are included in this article.

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
