# Peer review of "Sitagliptin Induces Tolerogenic Human Dendritic Cells"

_ijms, 2023, doi:10.3390/ijms242316829_

Round 1
Reviewer 1 Report
Comments and Suggestions for Authors
This manuscript reports the effect of the compound sitagliptin on tolerogenic and pro-inflammatory features of human monocyte-derived dendritic cells (moDCs) using basically standard methodology in the field. The experiments and methods seem to have been competently executed. The regulation of moDC tolerogenicity and subsequent lymphocyte stimulation is a very important issue for more potent DC-based immunotherapy and with several aspects still in need of clarification. The study seems experimentally sound and the role of sitagliptin is new. Publication is therefore warranted and recommended.
Specific points for revision:
It is unclear how MoDC medium was replenished, e.g. at Day 4: Was LPS and IFNG simply added to the mMoDC cultures or was medium changed with new sitagliptin (+/-) for both mMoDC and ImMoDC cultures? Could a schematic figure, e.g. in Supplemental materials, be useful to show the timeline of culture conditions both for the DC differentiation and maturation and for the allogeneic MLR?
Culture conditions are not well defined for the MLR, which medium and supplements?
Why is the order of coloured bars different in Figure 7A, for ILT3 (wrong colour or order?)? Sth. wrong with IDO-1 bar for imMoDC in the same Figure?
Does the chosen standard concentration of sitagliptin (500 microgram per ml) show beginning apoptosis and necrosis (Figure 1), i.e. that experiments could work “on the edge”? Likewise, there might be a problem with the relevance of this culture concentration of sitagliptin with reference to achievable in vivo concentrations. I am, however, willing to accept that this objection is addressed adequately in the Discussion part.
The complete Wb images were not submitted, only the focused mw regions, as also shown in Fig. 9B.
Author Response
Reviewer 1
Comments and Suggestions for Authors
This manuscript reports the effect of the compound sitagliptin on tolerogenic and pro-inflammatory features of human monocyte-derived dendritic cells (moDCs) using basically standard methodology in the field. The experiments and methods seem to have been competently executed. The regulation of moDC tolerogenicity and subsequent lymphocyte stimulation is a very important issue for more potent DC-based immunotherapy and with several aspects still in need of clarification. The study seems experimentally sound and the role of sitagliptin is new. Publication is therefore warranted and recommended.
Specific points for revision:
It is unclear how MoDC medium was replenished, e.g. at Day 4: Was LPS and IFNG simply added to the mMoDC cultures or was medium changed with new sitagliptin (+/-) for both mMoDC and ImMoDC cultures? Could a schematic figure, e.g. in Supplemental materials, be useful to show the timeline of culture conditions both for the DC differentiation and maturation and for the allogeneic MLR?
REPLY
We modified the methodology regarding this question. The corrections are marked in red (lines 520-525; 529-533; 537-538). A schematic presentation of the whole process of differentiation and maturation of MoDCs (control, sita 0d, and sita 4d protocols) and MoDCs/T-cell co-culture protocols is given in Supplementary Figure 4, as suggested.
Culture conditions are not well defined for the MLR, which medium and supplements?
REPLY
This is added in the Methodology section (see lines. 586-587; 546-548).
Why is the order of coloured bars different in Figure 7A, for ILT3 (wrong colour or order?)? Sth. wrong with IDO-1 bar for imMoDC in the same Figure?
REPLY
Sorry for this mistake, this is now corrected (see new Figure 8)
Does the chosen standard concentration of sitagliptin (500 microgram per ml) show beginning apoptosis and necrosis (Figure 1), i.e. that experiments could work “on the edge”? Likewise, there might be a problem with the relevance of this culture concentration of sitagliptin with reference to achievable in vivo concentrations. I am, however, willing to accept that this objection is addressed adequately in the Discussion part.
REPLY
Although this might be hypothesized based on the presented histogram, actually, statistical data did not confirm such an opinion. In one donor apoptosis was even lower than in the control. Additional arguments are presented in Supplementary Table 2 which shows the viability data after the whole cultivation period (described also textually in lines 146-149; 573-577). We agreed that this concentration is high (as already discussed), but there are clinical data about six times higher concentrations for the prevention of alloreactive transplantation reaction, without significant adverse effects. In addition upon request of another reviewer, we presented preliminary data about lower concentrations of sitagliptin (100 and 250 µg/mL, Supplementary Table 1). We also clarified this issue at the end of the discussion (lines 497-499).
The complete Wb images were not submitted, only the focused mw regions, as also shown in Fig. 9B.
REPLY
We apologize for a shorter description of this part in the Material and Methods section. As you can see from this extended description (lines 663-668) we sent the original strips as a valid document. The description is as follows: The proteins isolated from samples as well as the protein ladder (BlueStar Prestained Protein Marker, MWP03) were transferred to the nitrocellulose membrane. According to the appropriate ladder bands, the membrane was divided into strips spanning regions around the targeted protein, between 53kD and 93kD for p65/pp65, and between 23kD and 41kD for p38/pp38. The membrane strips were further separately incubated with appropriate antibodies. Expression of total p65 (p65), total p38, and activated form of p65 (phosphorylated p65, pp65) and activated form of p38 (pp38) was quantified by Western blot. The fold changes of pp65 (pp38) in comparison to total p65 (p38) in each sample were calculated and compared between the experimental groups. A similar approach was used in different articles (10.1177/1753425915586075, 10.1093/toxsci/kfm178, (our reference 62, which is also cited in the methodology section), 10.1371/journal.pone.0029155, 10.1016/j.jacc.2011.07.053).
Reviewer 2 Report
Comments and Suggestions for Authors
There are some typos in the manuscript and minor edits are required.
Reviewer 3 Report
Comments and Suggestions for Authors
The paper on "Sitagliptin induce toleranceogenic human dendritic cells" is a paper on the efficacy of human dendritic cells in sitagliptin and their regulatory mechanisms. Various experiments have been done, however, there are some experiments and figures that need to be supplemented.
1. Overall, you should be kind in explaining abbreviations in the manuscript.
2. Cytotoxicity is observed in high concentrations of Sita. The concentration of the sita used in the test shall be specified in all figures.
3. Corrections are required for the histogram that quantifies the figures in FACS.
4. In Figure 3, when the imMoDC and mMoDC differentiated and when the sita was treated should be clearly written, and the survival rate of DC should also be shown in 4d after the sita is treated.
5. It is necessary to explain CD26 observed in Figure 4 and the tolerance of human dendritic cells described in the paper.
6. Figure 5 requires a detailed explanation of cytokine productivity efficacy for sita.
7. The histogram is too large in the overall figure. It would be nice to see it again in the arrangement of the image so that the reader can see it easily.
8. All FACS data labels are clearly visible.
9. There is a need for a positive control that can quantify the Western blot result in Figure 9B.
Comments on the Quality of English Language
Overall, there is no problem with English. I just want you to check again.
Author Response
Reviewer 3
The paper on "Sitagliptin induce toleranceogenic human dendritic cells" is a paper on the efficacy of human dendritic cells in sitagliptin and their regulatory mechanisms. Various experiments have been done, however, there are some experiments and figures that need to be supplemented.
1.Overall, you should be kind in explaining abbreviations in the manuscript.
REPLY
The list of abbreviations is added at the end of the manuscript.
- Cytotoxicity is observed in high concentrations of Sita. The concentration of the sita used in the test shall be specified in all figures.
REPLY
Corrected as suggested
- Corrections are required for the histogram that quantifies the figures in FACS.
REPLY
All corrections are made and explained in detail for questions 6 and 7.
- In Figure 3, when the imMoDC and mMoDC differentiated and when the sita was treated should be clearly written, and the survival rate of DC should also be shown in 4d after the sita is treated.
REPLY
This issue was clarified in Materials and Methods (see lines 520-525; 529-533; 537-538). The data about the viability of MoDCs (all settings), studied at day 5 (the end of differentiation/maturation period), including im- and mMoDCs are presented in Supplementary Table 2 and explained in the text (lines 146-149; 573-577).
- It is necessary to explain CD26 observed in Figure 4 and the tolerance of human dendritic cells described in the paper.
REPLY
The significance of the increased expression of CD26 and its possible role in dendritic cells has been already discussed. It is not known, without new experiments whether CD26, itself, plays a role in tolerogenic functions of DC in sitagliptin-treatment protocols, this is now mentioned at the end of the discussion (see lines 501-503).
- Figure 5 requires a detailed explanation of cytokine productivity efficacy for sita.
REPLY
We rewrote some textual parts of the results regarding this topic, so now the explanation is more clear (lines 177-187).
- The histogram is too large in the overall figure. It would be nice to see it again in the arrangement of the image so that the reader can see it easily.
REPLY
Since there is no limitation in the number of Figures, we split some Figures and arranged the plots to be more visible and also enlarged the numbers on some Figures. Therefore the total number of Figures is now 12.
- All FACS data labels are clearly visible.
- There is a need for a positive control that can quantify the Western blot result in Figure 9B.
REPLY
We apologize for the shorter description of this part in the Material and Methods section. This is now corrected (664-669; 681-682). The expression of total p65 (p65), total p38, and activated p65 (phosphorylated- pp65) and activated p38 (pp38) was quantified by Western blot. The fold change of pp65 (pp38) in comparison to total p65 (p38) in each sample was calculated and compared between the groups. NFκB and p38 MAPK are central mediators of priming and activating signals, such as different pro-inflammatory cytokines (IFN-γ) and PAMPs (LPS), in immune cells, that activate the transcription of target genes involved in inflammation development and progression (doi.org/10.1038/sigtrans.2017.23, 10.4049/jimmunol.1302700). Taking in mind this, the sample from MoDC stimulated only with LPS/ IFN-γ could be considered also as a positive control. Total p65 (p38) was also used as a loading control which is a frequently used approach in these analyses (10.1177/1753425915586075, 10.1093/toxsci/kfm178, 10.1371/journal.pone.0029155, 10.1016/j.jacc.2011.07.053).
Round 2
Reviewer 2 Report
Comments and Suggestions for Authors
The authors carefully addressed my concerns and recommendations.
Author Response
No comments.
Reviewer 3 Report
Comments and Suggestions for Authors
The author has revised it well according to the comments, but there are results that need to be supplemented.
1. In Figure 12, although the total form of p65 and p38 was confirmed, b-actin or GAPDH must be present as a loading control.
